# CLASS-SPECIFIC FEATURE LEARNING THROUGH MITIGATING SPURIOUS CORRELATION IN CONTEXT-POOR CLASSES FOR LONG-TAILED CLASSIFICATION

## ABSTRACT

The inevitably long-tailed data in real-world presents a challenge for deep learning methods. Imbalanced data causes the context in each class to be unevenly distributed, prompting models to prioritize the accurate classification of context-rich classes while largely disregarding context-poor classes. Firstly, from both theoretical and practical perspectives, we explain why the model tends to favor the context-rich classes from both training and test sets, and breaking the spurious correlation between class-specific and context-poor features will further improve the effectiveness and robustness. Then, we propose a framework termed spurious correlation of context-poor, which aims to focus on class-specific features by progressively breaking the spurious correlation of limited data. Specifically, Grad-CAM is utilized to segment contextual regions and foregrounds within the samples coarsely. The high-confidence masks are retained and used to generate samples for the context-poor classes. Subsequently, to further mitigate spurious correlation, more reasonable class centers are incorporated into the contrastive loss to minimize the distance between semantically similar samples while maximizing the separation between samples from different classes in the feature space. To better preserve discriminative features, supervision is also performed on the generated samples. Finally, the experiments conducted on CIFAR10/100-LT, iNaturalist 2018, and ImageNet-LT demonstrate the effectiveness of our model. The code is available in the supplementary material.

## 1 INTRODUCTION

Data in real world always show long-tailed distributions which mean that the few head classes occupy a large number of training samples and the many classes (tail classes) are under-represented Wang et al. (2021b); Chen et al. (2023). Such a severely unbalanced distribution proposes new challenges for traditional classifiers Alshammari et al. (2022); Li et al. (2022). In long-tailed distribution, either head classes or tail classes, there is also another imbalance in each class, i.e., context imbalance. Compared to medium and tail classes, head classes have more training samples with various context. As a result, context-poor classes are more likely to appear among the medium and tail classes. Suppose we have a sufficient number of training samples in each class with different backgrounds, then only the traditional classifier is needed, and we can also achieve a better performance. However, long-tail distribution will inevitably in real world, and the solution to this imbalance problem is expensive Zhang et al. (2023). Currently, the mainstream methods utilize the class imbalance ratio to rebalance tail and head classes confidences, expecting to obtain a classifier that is unbiased to the head classes but pay more attention to the tail classes Ren et al. (2020); Menon et al. (2021).

As depicted in Figure 1, the spurious correlation is caused by the fact that some rare backgrounds are incorrectly targeted to certain classes. Models pay more attention to the context rather than the discriminative features of class, such as those rare backgrounds: sky and grass. The models tend to strongly bind to rare context features to reduce training costs, i.e., overfitting. The reason is that the affluent and diversified context information of head classes can be utilized in the training, so that enough discriminative features of class can be learned and the spurious correlation are weaker. Compared to the head classes, the tail and medium classes are data-limited and information-limited, which makes the model tend to overfit the features of tail and medium

classes, especially for context-poor classes, resulting in a strong spurious correlation. Therefore, a natural question is: Can mitigating the spurious correlation benefit context-poor classes?

To verify this issue, we first utilize Grad-CAM Selvaraju et al. (2017) to generate heatmaps to validate the attention of models, thereby further proving the spurious correlation between the context-poor features and the discriminative features of the class. The hotmap of context-poor/rich class (redshank/indigo bird) is showed in Figure 1. For the head classes with information-rich and context-rich data, both the CE-based model and our method are capable of capturing the core features of partridge. In contrast, for context-poor class redshank, in uncommon background (e.g., grasslands), although the CE-based model also focuses on the samples's outline. Specifically, as depicted in Figure 1(a), it tends to emphasize context features (red regions) such as grassland and sky rather than class-specific attributes (e.g., slender legs and beak). It will lead the model to learn context-dependent features instead of invariant class-specific features, resulting in a spurious correlation between context and class-specific features. Thereby impairing the ability to distinguish tail class samples of model. As depicted in Figure 1(b), our method pay more attention to the class-specific features rather than irrelevant context features, effectively decoupling the core and context features and mitigating the spurious correlation.

Figure 1 also provides experimental evidence of this issue. We trained 200 epochs for resnet-32 He et al. (2016) with CE loss on CIFAR100 with various imbalance ratios (e.g., 200, 100 and 50). From the results, we can see that the model is effective in the training set whether in head, medium or tail classes. As shown in the test set of Figure 1, with the in-

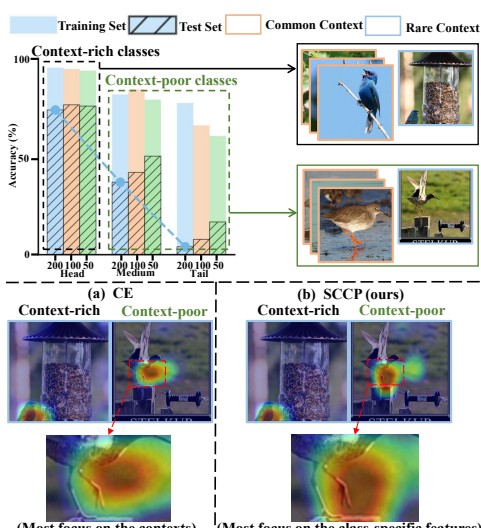

Figure 1: The explanation illustrates the spurious correlation between class-specific features and context features in context-poor classes. The tail classes exhibit a larger training/test set accuracy gap, indicating severe spurious correlation. The red regions denote higher model attention, while lighter colors represent lower levels of attention. In (a) and (b), for the classes with diverse and balanced context, both the CE and SCCP are capable of focusing on class-specific features. For the context-poor classes, CE predominantly focuses on context features, whereas SCCP primarily emphasizes class-specific features.

crease of imbalance ratio, the degree of overfitting of class is also constantly increasing, especially for the tail classes, this means that spurious correlation become more severe as the imbalance ratio increases. To elaborate on this issue, we take CIFAR100 with IR 200 as an example. As illustrated in the line graph in Figure 1, the performance of tail classes decreases significantly compared to that of head classes. Experiments confirm that the model can not learn discriminative feature representations of tail classes, which also verifies the hypothesis we proposed above: spurious correlation.

We posit that context-poor features across head, medium, and tail classes may cause the model highly semantically related to the irrelevant context features, and the spurious correlation between class-specific and context features is formed. Eliminating such spurious correlation could significantly benefit long-tail representation learning. The key point to improving classification performance for rare samples depend on breaking the spurious correlation between context and class-specific features while preserving the essential discriminative features of each class. Therefore, we propose a framework termed Spurious Correlation of Context-Poor (SCCP) to enhance the model's generalization by progressively learning from insufficient data. Firstly, Grad-CAM is utilized to roughly segment the contextual regions and foregrounds of sample, which are then preserve the mask with high confidence among them and used to generate new synthetic samples. Subsequently, more reasonable class centers are adopted to contrastive loss to narrow the gap of similar samples and increase the distance between the samples of different classes more reasonably, thereby effectively breaking spurious correlation. Finally, the supervision is performed on the generated samples to ensure that the class-specific features of classes can be effectively distinguished. In experiments, empirical results are conducted on four long-tailed datasets: CIFAR10/100-LT Cao et al. (2019),

iNaturalist 2018 Van Horn et al. (2018), and ImageNet-LT Liu et al. (2019). The results demonstrate that SCCP achieves competitive performance compared to current state-of-the-art algorithms. Furthermore, experiments and ablation studies can demonstrate aforementioned discovery and the effectiveness of SCCP. Our main contributions are as follows:

- We conduct comprehensive both theoretical and empirical analysis on different datasets to investigate the spurious correlation between rare context features and class-specific features. Our findings suggest that such correlation are particularly severe in context-poor classes, primarily attributable to the insufficient contextual diversity of context-poor classes.

- We propose a structural approach designed to mitigate the spurious correlation between context features and class-specific features in context-poor classes (i.e., tail or medium classes), while retaining the discriminative features of the context-poor classes which can exhaustively boost the representation ability of the model.

- We evaluate the proposed model on four long-tailed datasets (e.g., CIFAR10-LT, CIFAR100-LT, iNaturalist 2018 and ImageNet-LT). The effectiveness of our method is demonstrated by extensive experiments and ablation studies, the proposed method all achieves competitive results in diverse settings.

## 2 RELATED WORK

The related work is presented in Appendix C.

## 3 METHODOLOGY

### 3.1 PROBLEM FORMULATION

Following the previous methods, the classification model is formulated as $p(y|x)$, which utilizes the input $x$ to predict the label $y$. After Bayes expansion, $p(y|x) \propto p(x|y) \cdot p(y)$. The validity of this formula relies on the strong assumption that $p_{tr}(x|y) = p_{te}(x|y)$. However, this assumption cannot be guaranteed in practical applications.

In prior studies Tang et al. (2022); Qi et al. (2022), it is generally assumed that an image $x$ comprises both context and class-specific features; however, existing methods often overlook the presence of spurious correlation features, a problem that is especially pronounced in long-tail learning scenarios. Therefore, each image can be represented by a set $(f_{cs}, f_{ct}, f_{sc})$, where $f_{cs}$ denotes the class-specific features that capture the core attributes of the category, and $f_{ct}$ represents the context features corresponding to background information surrounding the target. While these features can facilitate classification, excessive reliance on $f_{ct}$ may contribute to spurious correlation, denoted as $f_{sc}$. In contrast to image captioning and segmentation tesks, which rely on comprehensive image features, classification model are concerned primarily with class-specific and the relevant context features. Therefore, we expand the $p(y|x)$ to $p(y|f_{cs}, f_{ct}, f_{sc})$, which only assumes the class-specific features $f_{cs}$ is invariant, i.e., $p_{tr}(f_{cs}|y) = p_{te}(f_{cs}|y)$. Follow the Bayes rule, we arrive at

$$p(y|f_{cs}, f_{ct}, f_{sc}) = \underbrace{\frac{p(f_{cs}|y)}{p(f_{cs})}}_{\text{class-specific}} \cdot \underbrace{\frac{p(f_{ct}|y)}{p(f_{ct})}}_{\text{relevant-context}} \cdot \underbrace{\frac{p(f_{ct}, f_{cs})}{p(f_{ct}|y)} \cdot \frac{p(f_{ct})}{p(f_{ct}|f_{cs})} \cdot \frac{p(f_{sc}|y, f_{cs}, f_{ct})}{p(f_{sc}|f_{cs}, f_{ct})}}_{\text{spurious correlation}} \cdot p(y). \quad (1)$$

In the Eq. 1, the class-specific component $f_{cs}$ only depend on $y$; the relevant-context component (e.g., the desert to camels and the water surface to waterfowls) which can facilitate classification also only depend on $y$. To enable effective classification, the model should effectively distinguish class-specific and context features, while still retaining and using contextual information rather than completely discarding it. That is to say, a well-designed model is generally expected to learn not only class-specific features but also relevant context features that serve as auxiliary information, while avoiding excessive reliance on such context features $f_{ct}$. Based on this, it naturally arrives at the following assumption:

**Assumption 1** Given the class label $y$, the class-specific features $f_{cs}$ is independent on the context features $f_{ct}$, i.e., $f_{cs} \perp f_{ct}|y$.

Given this assumption mentioned above, the Eq. 1 can be simplified as

$$p(y|f_{cs}, f_{ct}, f_{sc}) = \underbrace{\frac{p(f_{cs}|y)}{p(f_{cs})}}_{\text{class-specific}} \cdot \underbrace{\frac{p(f_{ct}|y)}{p(f_{ct})}}_{\text{relevant-context}} \cdot \underbrace{\frac{p(f_{ct})}{p(f_{ct}|f_{cs})} \cdot \frac{p(f_{sc}|y, f_{cs}, f_{ct})}{p(f_{sc}|f_{cs}, f_{ct})}}_{\text{spurious correlation}} \cdot p(y). \qquad (2)$$

The spurious correlation component is consists of two components. $p(f_{sc}|y, f_{cs}, f_{ct})/p(f_{sc}|f_{cs}, f_{ct})$ depends on $f_{cs}, f_{ct}, y$, while $p(f_{ct})/p(f_{ct}|f_{cs})$ quantifies the correlation between class-specific features and context features. The value close to 1 suggests that the context features are utilized to aid classification without becoming dominant. In contrast, a deviation from 1 indicates that class-specific and context features are entangled, thereby giving rise to spurious correlation. The detailed derivation process of Eq.1 and 2 are presented in the Appendix A.

Different from Tang et al. (2022); Qi et al. (2022), we posit that both class-specific and context features contribute to enhancing the model's classification performance. However, excessive reliance on context features may induce spurious correlation. As illustrated in Figure 1, this issue is particularly pronounced for tail classes samples that possess rare contextual information. Therefore, it is necessary to break the spurious correlation between class-specific and context features, thereby effectively enhancing the model's classification performance.

Meanwhile, we further conducted an analysis of this issue from the perspective of loss. As illustrated in Figure 2, taking CIFAR-100 with imbalance ratio 100 as an example, the CE loss distribution of the head, medium, and tail classes are presented for both training and test sets. It can be observed that the loss distribution of the training and test sets gradually separate from head to tail classes. This indicates a significant issue: for the traditional model, the loss distribution of head classes on the training and test sets is unified, suggesting that the model performs well in identifying these classes in training and test phases. In contrast, for the tail classes (context-poor

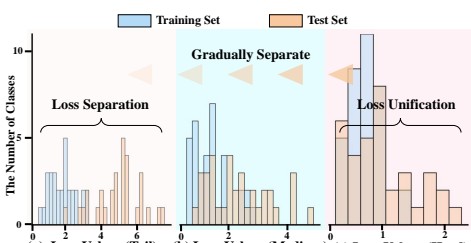

Figure 2: The loss separation of training and test sets indicates that traditional model is ineffective in tail classes.

classes), the loss distribution of the training and test sets are almost completely separated, indicating that the model treats some test set samples of tail classes as noises or outliers. This discovery further indicates that the traditional models often entangle class-specific and context features, thereby inducing spurious correlation and overfitting, particularly for context-poor classes. Therefore, we infer that if the spurious correlation of context-poor classes can be breaked, the performance can approach that of the context-rich classes.

## 3.2 MITIGATING THE SPURIOUS CORRELATION OF THE CONTEXT-POOR CLASSES

Motivated by the aforementioned analysis, we propose a method termed Breaking Spurious Correlation (BSC) that can preserve the discriminative features of each class and effectively mitigate the spurious correlation. The mentioned above research reveals that the tail classes have insufficient context information. Since the scarcity of the context features, the models tend to overfit these rare context in order to achieve better training performance. This behavior significantly undermines the generalization capability of the models. Meanwhile, the model has demonstrated outstanding classification performance in the common context, whether it is the head classes or tail classes. Moreover, the rare context within the head classes can also be accurately classified due to the abundance of context information available, which facilitates effective learning of class-specific features. Therefore, we conclude that models tend to overfit the features of context-poor classes, and enhancing the context features of the tail classes can improve the model's ability to learn discriminative feature representations.

To prevent overfitting to the features of context-poor classes with scarce data, we exploit the abundant context from other classes to enhance context-poor classes, thereby improving the performance of context-poor classes. We utilize the Grad-CAM to mask the foreground and background regions, allowing us to extract context information from the tail and other classes. Given the input samples

$(x_{iter}, y_{iter})$, the new samples $(\widetilde{x}_i, \widetilde{y}_i)$ is formalized in Eq. (3).

$$\widetilde{x}_i = M_i \odot x^i_{iter} + (1 - M_i) \odot x^i_{bank},$$

$$\widetilde{y}_i = y^i_{iter},$$

$$M_i = \begin{cases} 1, & mask^i_{iter} = 1, 0.5 \leq S < 0.8 \\ mask^i_{bank}, & mask^i_{iter} = 0 \end{cases}, \quad (3)$$

$$mask_{(j,k)} = \begin{cases} 1, & mask_{(j,k)} \geq 0.5 \\ 0, & mask_{(j,k)} < 0.5 \end{cases},$$

where $x_{iter}$ and $x_{bank}$ denote the samples from current iteration and the samples with high-confidence score saved from previous iterations, respectively. $mask_{j,k}$ represents the foreground-background distribution mask generated using Grad-CAM, which are used to extract corresponding regions from the image. If the value of $mask_{j,k}$ is greater than or equal to 0.5, it is defined as foreground regions, conversely, it is recorded as the background regions. Meanwhile, the ablation studies can also demonstrate the effectiveness of the value. $S$ indicates the confidence score of the samples in the current iteration. For sample with low-confidence score, the Grad-CAM mask may not be reliable; for those with high-confidence scores, frequent alteration of the context information of samples may affect the model to learn class-specific feature representations. Therefore, samples with low-confidence or high-confidence scores are not suitable, we only utilize samples with confidence scores between 0.5 and 0.8 to generate new samples. Given that most samples stored in the $x_{bank}$ originate from head classes, to prevent the excessive class-specific features of head classes that could interfere the learning of tail classes. Therefore, we constraint the $mask_{j,k}$ to obtain $M_i$. To be specific, the binary step function is applied to the $mask_{j,k}$, setting the value to either 0 or 1, instead of directly blending $x^i_{iter}$ and $x^i_{bank}$ to generate new samples. Furthermore, for the background regions of $x^i_{iter}$, the foreground of $x^i_{bank}$ is also not extracted. It effectively preserves the discriminative feature of context-poor classes while enriching their contextual diversity.

The capacity of $x_{bank}$ equal to the batch size of iteration, and $x_{bank}$ is updated in each iteration. Only samples with a confidence score greater than 0.8 are stored in $x_{bank}$. The majority of samples belong to head classes in the long-tail distribution, therefore, the samples with high-confidence score in $x_{bank}$ almost from the head classes in the early stage. This phenomenon also shown in the classification results of the training set presented in Figure 1, this leads to $x_{bank}$ biased to the head classes. To enhance contextual diversity, it is necessary to retain as many classes as possible in $x_{bank}$. To achieve this goal, we implement a retention strategy for context-poor classes. Specifically, if a class has only one sample stored in $x_{bank}$, it won't be removed even when $x_{bank}$ is full. For classes with more than one stored sample, the samples that exceed the capacity will be deleted proportionally. The number of samples to be removed from each class is computed as follows:

$$Del^c_{Num} = \begin{cases} Del_{Num} * Del^c_{Pro}, & Num^c > 1 \\ 0, & Num^c = 1 \end{cases}, \quad (4)$$

where $Del_{Num}$ denotes the number of samples exceeding the capacity of $x_{bank}$, $Del^c_{Num}$ represents the number of samples in class $c$ should be deleted, $Del_{Pro}$ represents the proportion of classes in the $x_{bank}$, and $Num_c$ denotes the number of samples belonging to class $c$ within the $x_{bank}$.

### 3.3 THE LEARNING FRAMEWORK OF MITIGATING THE SPURIOUS CORRELATION OF THE CONTEXT-POOR CLASSES

For the context-poor classes (e.g., tail classes), due to the limited availability of data, it becomes challenging to effectively capture the core features of these categories. This scarcity leads the model to overfit the rare-context features. Preserving the discriminative features of context-poor classes while mitigating spurious correlation with context features is a key factor in improving performance.

To address this issue, we first use the Grad-CAM to generate sample $(\tilde{x}, \tilde{y})$, both the generated samples and the samples from the current iteration are utilized to update the class centers $(W_1, ..., W_N)$. Then, more reasonable class centers are adopted to contrastive loss for reducing the intra-class distance in the feature space while increasing the inter-class separation. By introducing samples $(\tilde{x}, \tilde{y})$, the contextual diversity of underrepresented categories is enhanced, which facilitates the formation of more generalized class centers $(W_1, ..., W_N)$ and reduces the susceptibility to spurious correlation

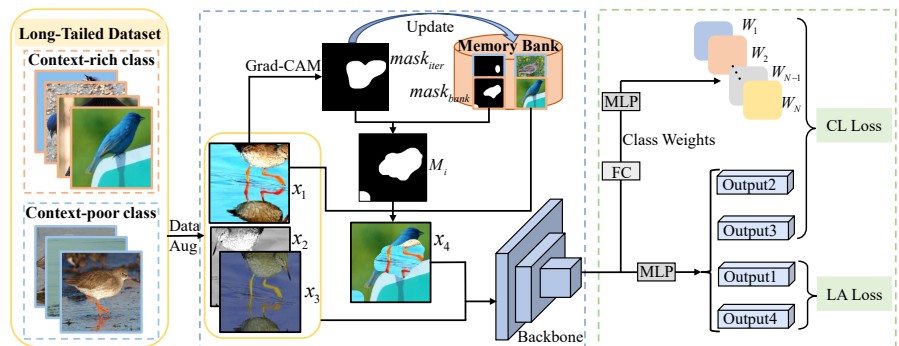

Figure 3: The overview of the proposed framework.

and overfitting. The analysis in Figure 4 further confirm this conclusion. Secondly, we posit that the model should further enhance its ability to learn discriminative class features, thereby improving its classification performance on samples with diverse context variations. To this end, we apply the LA loss Menon et al. (2021) to supervise the generated samples with invariant class-specific and varied context information to reinforce the models capacity of capturing class-specific features. This strategy mitigates the model's bias toward context-rich classes, further enhancing its ability to capture discriminative class features and break the spurious correlation of context features. The learning framework of breaking the spurious correlation of the context-poor classes illustrate in Figure 3.

As depicted in Figure 3, the proposed framework first applies BSC to decouple rare contextual dependencies, thereby mitigating the spurious correlation. And then, we derive some more reasonably semantically class centers $(W_1, ..., W_N)$ to enrich the generalization of context-poor classes, and the supervised training is utilized to the generated samples $(\tilde{x}, \tilde{y})$, thereby enhancing the discriminative ability of the model with respect to class-specific features. The loss function is defined as follows:

$$\mathcal{L} = \mathcal{L}_{CL}(O_2, O_3, C, T) + \mathcal{L}_{LA}(O_1, T) + \mathcal{L}_{LA}(O_4, T), \tag{5}$$

where $\mathcal{L}_{CL}$ and $\mathcal{L}_{LA}$ denote contrastive loss and LA loss, $O$ and $C$ denote outputs and class centers.

**Mechanisms behind the SCCP.** We also conduct experiments to explore how SCCP enhances the performance of classifier. We present the benefits from three perspectives: (1) representation analysis, (2) performance improvements and (3) clustering analysis, the results are shown in Figure 4. We adopt t-SNE Visualization Maaten & Hinton (2008) to visualize the learned feature representations in a 2D space. Compared with the BCL, the t-SNE visualization of SCCP reveals well-separated clusters and a discriminative structure corresponding to different classes. This demon-

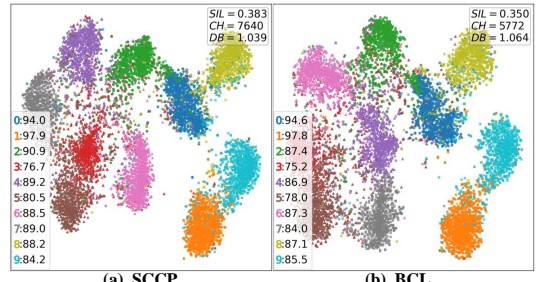

Figure 4: The t-SNE visualization and cluster analysis for model validity interpretation.

strates that SCCP is more effective in capturing discriminative representations. Meanwhile, considering the classification performance of each class, SCCP attained the best results in 8 out of 10 classes. Finally, to further verify this conclusion, we used three clustering evaluation metrics (e.g.,Silhouette Coefficient (SIL), CalinskiHarabasz score (CH), and DaviesBouldin score (DB) score Rousseeuw (1987); Caliński & Harabasz (1974)) to provide the corresponding quantitative analysis results. Both the DB and the SIL are used to evaluate the intra-cluster compactness and inter-cluster separation in clustering results. Lower values for DB and larger values for SIL indicate more cohesive and better-separated clusters. The CH quantifies the ratio of between-cluster dispersion to within-cluster dispersion, with higher values reflecting better-defined and more separable cluster structures. Based on these experimental results, SCCP consistently outperforms the BCL. This demonstrates that our model achieves excellent intra-class compactness and increased inter-class separability within the feature space, thereby exhibiting superior classification discriminability and robustness. Furthermore, these findings suggest that SCCP is capable of learning sufficient class-specific features while simultaneously leveraging context features for auxiliary classification,

without exhibiting excessive dependence on them. As a result, it suppresses spurious correlation and yields improved model performance.

## 4 EXPERIMENTS

### 4.1 MAIN RESULTS ON CIFAR10/100-LT, INATURALIST 2018 AND IMAGENET-LT

Implementation details is provided in Appendix B.1, the highest-performing results are in bold, ∗ denotes the results quoted from corresponding papers Shao et al. (2024); Zhou et al. (2023), the results of the remained methods are reported from the original paper, entries not presented in the original papers or the corresponding references are indicated as "-".

**Experimental Results on CIFAR10/100-LT.** The experimental results of CIFAR10-LT and CIFAR100-LT are reported in columns 3-8 of Table 1. For a fair comparison, we compare the performance of different methods with various imbalance ratio (e.g., 200, 100, and 50) on the CIFAR10/100-LT datasets. As shown in the Table 1, the detailed results show that SCCP achieves the superior accuracy in all settings. To be specific, our method surpasses the other algorithms, achieving hightes accuracy of 82.0%, 85.5%, and 87.9% for CIFAR10-LT and 47.8%, 52.6%, and 56.8% for CIFAR100-LT, respectively. The proposed method at least enhances accuracy by 2.6%, 0.8%, and 1.0% on CIFAR10-LT and by 1.3%, 0.7%, and 0.2% on CIFAR100-LT across the various imbalance ratios, further validating the effectiveness of SCCP. As shown in the Figure 4, since SCCP breaks the spurious correlation between context and class-specific features in context-poor classes, they can be easily extract the discriminative features, wherein intra-classes exhibit greater concentration and inter-classes demonstrate enhanced separation. The detailed experimental results (e.g., head, medium, and tail classes) of CIFAR100-LT are presented in the Appendix B.2.

**Experimental Results on large-scale datasets: iNaturalist 2018 and ImageNet-LT.** The experi-

Table 1: The classification accuracy of CIFAR10/100-LT, iNaturalist 2018 and ImageNet-LT.

| Dataset | | CIFAR10 | | | CIFAR100 | | | iNa-18 | Ima-LT |
|---|---|---|---|---|---|---|---|---|---|
| Method | Reference | 200 | 100 | 50 | 200 | 100 | 50 | 256 | 512 |
| LDAM∗ Cao et al. (2019) | NeurIPS19 | - | 77.0 | - | - | 42.0 | - | 70.1 | 52.9 |
| CBL Cui et al. (2019) | CVPR19 | 68.9 | 74.6 | 79.3 | 36.2 | 39.6 | 45.3 | - | - |
| Decouple∗Kang et al. (2019) | ICLR20 | - | - | - | - | - | - | 65.6 | 46.7 |
| BS∗ Ren et al. (2020) | NeurIPS20 | - | - | - | - | - | - | 70.6 | 52.3 |
| MCW Jamal et al. (2020) | CVPR20 | 68.9 | 75.2 | 80.1 | 37.9 | 42.1 | 46.7 | - | - |
| LA Menon et al. (2021) | ICLR21 | - | 77.7 | - | - | 43.9 | - | 66.4 | 51.1 |
| IB Park et al. (2021) | ICCV21 | 74.0 | 78.3 | 81.7 | 37.3 | 42.1 | 46.2 | - | - |
| LADE∗ Hong et al. (2021) | CVPR21 | - | - | - | - | - | - | 70.0 | 52.3 |
| RSG∗ Wang et al. (2021a) | CVPR21 | - | - | - | - | - | - | 70.3 | 51.8 |
| DisAlign∗ Zhang et al. (2021) | CVPR21 | - | - | - | - | - | - | 70.6 | 52.9 |
| MiSLAS∗ Zhong et al. (2021) | CVPR21 | - | - | - | - | 47.0 | 52.3 | 71.6 | 52.7 |
| GCL Mengke Li (2022) | CVPR22 | 76.6 | 81.3 | 85.2 | 43.0 | 47.4 | 51.4 | 72.0 | 54.9 |
| ResLT Cui et al. (2022) | TPAMI22 | - | 80.4 | 83.5 | - | 45.3 | 50.0 | 70.2 | 52.9 |
| BCL Zhu et al. (2022) | CVPR22 | - | 84.3 | 87.2 | - | 51.9 | 56.6 | 71.8 | 56.0 |
| CMO Park et al. (2022) | CVPR22 | - | - | - | - | 47.2 | 51.7 | - | - |
| SAM∗Rangwani et al. (2022) | NeurIPS22 | - | 81.9 | - | - | 45.4 | - | - | - |
| CSA Shi et al. (2023) | NeurIPS23 | - | 82.5 | 86.0 | - | 46.6 | 51.9 | - | 49.7 |
| ADRW∗ Wang et al. (2023) | NeurIPS23 | - | 83.6 | - | - | 46.4 | - | - | - |
| AREA Chen et al. (2023) | ICCV23 | 75.0 | 78.0 | 82.7 | 43.9 | 48.8 | 51.8 | 68.4 | 49.5 |
| GCL-A Li et al. (2024a) | TAI24 | 79.3 | 82.7 | 85.6 | 46.5 | 50.0 | 54.8 | 71.1 | 55.1 |
| H2T Li et al. (2024b) | AAAI24 | - | - | - | 45.2 | 48.9 | 53.8 | 71.6 | 54.6 |
| DiffuLTShao et al. (2024) | NeurIPS24 | - | 84.7 | 86.9 | - | 51.5 | 56.3 | - | 56.4 |
| DisA Gao et al. (2024) | ICML24 | 79.1 | 82.8 | 84.9 | 45.2 | 49.8 | 54.4 | - | 54.5 |
| PLOT Zhou et al. (2023) | ICLR24 | 77.7 | 81.9 | 85.7 | 44.3 | 47.9 | 52.7 | 72.1 | 53.5 |
| SAM Li et al. (2025) | ICML25 | 79.6 | 82.9 | 85.5 | 46.0 | 50.7 | 54.5 | 71.8 | 54.3 |
| SCCP | Ours | **82.0** | **85.5** | **87.9** | **47.8** | **52.6** | **56.8** | **72.7** | **57.0** |

mental results of large scale datasets are presented in the last two columns of Table 1. It confirms the effectiveness of SCCP which can obtain a unbiased model by comprehensively extracting the discriminative feature representations and alleviating the spurious correlation between rare context and class-specific features in context-poor classes. The last two columns of Table 1 presents that the accuracy of our method on iNaturalist 2018 and ImageNet-LT, it shows that the proposed model is superior to most other methods. For iNaturalist 2018, SCCP achieves a 72.7% accuracy, outperforming the nearest method by 0.6%. On the ImageNet-LT, SCCP is improved to 57.0%, which is the best among all the comparative methods, and outperforms the second-best algorithm by 0.6%. It shows that SCCP can also better mitigate the spurious correlation and preserve discriminative class-specific features on large-scale datasets. We further compare the performance of SCCP on head, medium, and tail classes, the results is showed in the Appendix B.3.

## 4.2 VISUALIZATION

**Loss Landscapes Visualization**. The Figure 5 presents the loss landscapes visualizations Li et al. (2018); Rangwani et al. (2022) of entire CIFAR100 with imbalance ratio 100 in SCCP and BCL. It also demonstrates strong competitiveness of the proposed method. The visualizations of head, medium, and tail classes are present in Appendix B.4. From the Figure 5, we can observe that the loss surface of BCL is relatively low but highly sharp, suggesting that the model suffers from poor generalization performance and is prone to overfitting.

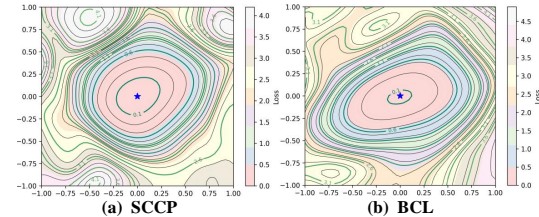

(a) SCCP  (b) BCL

Figure 5: The loss landscapes visualization of SCCP and BCL.

In contrast, the proposed method demonstrates lower loss values along with a more gradual and flat minima. This indicates that SCCP is capable of distinguishing class-specific and context features, thereby suppressing spurious correlation and having superior stability and generalization capability. Therefore, SCCP enables both the mitigation of saddle point entrapment and the enhancement of representational generalization. The results further support the conclusion that SCCP effectively mitigates the spurious correlation between context features and class-specific features.

**Hessian Spectrum Analysis**. To further illustrate the model's ability to break the spurious correlation between context and class-specific features, we also provide the hessian spectrum analysis Dinh et al. (2017); Dauphin et al. (2014); Rangwani et al. (2022) of CE, BCL and SCCP (corresponding to the three rows in the figure respectively) on CIFAR100 with long-tail ratio 100. The Figure 6 illustrates the largest and smallest eigenvalues ($\lambda_{max}$ and $\lambda_{min}$) across different groups (i.e., entire and tail, others is shown in the Appendix B.5). As the values of $\lambda_{max}$ and $\lambda_{min}$ approach zero, the greater the robustness of the model pair against interference, indicating excellent generalization performance and a reduced likelihood of overfitting. The proposed method results in flatter minima for the entire and tail classes of dataset. For the smallest and largest eigenvalues, our method achieved -2.3494, -9.8221 and 38.0715, 80.0646 across entire and tail classes, respectively. Compared with the CE and BCL, the $\lambda_{max}$ and $\lambda_{min}$ of SCCP across different

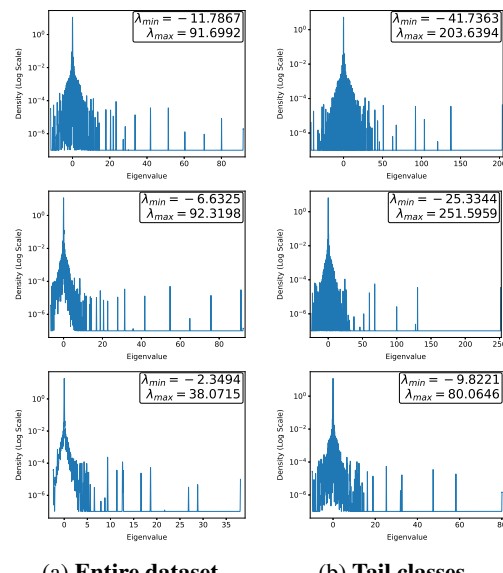

(a) **Entire dataset**   (b) **Tail classes**

Figure 6: The hessian spectrum analysis of CIFAR100-LT with imbalance ratio 100.

groups are more closer to zero, which indicates that our method mitigates the risk of being overfit and has more generalized representations. More analysis of each class, head, medium, and tail classes is available in the Appendix B.5.

### 4.3 ABLATION STUDY

Our method comprises multiple components and some hyper-parameters, to evaluate the effect of these components, we conduct ablation studies on CIFAR10/100-LT across various imbalance ratios to present the effectiveness of each proposed component, the performance is shown in Table 2-3. The ablation results indicate that the proposed operations can significantly enhance the classification performance. More ablation experiments can be found in the Appendix B.6.

**Different components in SCCP**. We analyze the effectiveness of each component on CIFAR10/100-LT spanning various imbalance ratios in Table 2. The rows 3-5 are the experimental results of CIFAR100-LT, and the rows 6-8 are the experimental results of CIFAR10-LT. Compared with the baseline, SCCP achieve the best results with an improvement of at least 1.8%, 2.7%, 2.2%, 1.2%, 2.1%, and 1.5% respectively. The results highlight that each component plays a crucial role in enhancing the overall classification performance. We further report the results of head, medium, and tail classes on CIFAR10/100-LT with imbalance ratio 100. While SCCP only dominates one tail classes on two datasets, its performance of head classes is similar to BSC, which means that it performs better in the overall medium and tail classes, i.e., context-poor classes. It indicates that the proposed method effectively alleviates the spurious correlation of context-poor classes.

**Different fusion methods in breaking spurious correlation**. To understand the effects of the different fusion methods in breaking the spurious correlation, we analyze several different fusion methods in Table 3. Specifically, four different fusion methods are presented in the Table 3, such as Random Fusion (RF), Mixup, Full Fusion (FF), and Retaining the Class-specific Features of other classes (RCF). As shown in the experimental results, our method improves the model performance obviously. Comapred with the Mixup, as the imbalance ratio increased, the improvement of SCCP also became greater, rising from 4.58% to 16.51% on CIFAR100-LT. Compared with the other methods, our algorithm has also been improved to a certain extent. Based on the analysis in the introduction, there exists a positive correlation between the imbalance ratio and the degree of spurious correlation. Therefore, the results show that our method can effectively alleviate this spurious correlation for various imbalance ratios.

Table 2: The ablation experiments for each proposed component in SCCP across various imbalance ratios (e.g., 200, 100, 50).

| Method | Imbalance Ratio | | | IR=100 | | |
|---|---|---|---|---|---|---|
| | 200 | 100 | 50 | Head | Medium | Tail |
| Baseline | 46.0 | 49.9 | 54.6 | 64.4 | 51.1 | 30.4 |
| BSC | 47.7 | 52.4 | 56.2 | 66.7 | **54.5** | 32.0 |
| SCCP | **47.8** | **52.6** | **56.8** | **66.9** | 53.5 | **33.9** |
| Baseline | 80.8 | 83.4 | 86.4 | 90.9 | 78.7 | 82.1 |
| BSC | 81.5 | 85.3 | 87.5 | **93.2** | 80.5 | **83.7** |
| SCCP | **82.0** | **85.5** | **87.9** | 92.7 | **83.3** | 81.1 |

Table 3: The ablation experiments to verify the impact of different fusion methods in breaking spurious correlation.

| Method | CIFAR100 | | | CIFAR10 | | |
|---|---|---|---|---|---|---|
| | 200 | 100 | 50 | 200 | 100 | 50 |
| Mixup | 31.2 | 49.4 | 53.2 | 79.7 | 83.5 | 85.6 |
| RF | 47.5 | 51.9 | 56.2 | 81.4 | 84.8 | 87.8 |
| FF | 46.8 | 51.4 | 56.0 | 81.6 | 85.0 | 87.2 |
| RCF | 47.7 | 52.1 | 56.6 | 81.7 | 85.2 | 87.3 |
| SCCP | **47.8** | **52.6** | **56.8** | **82.0** | **85.5** | **87.9** |

## 5 CONCLUSION

In this work, we investigate and analyze the problem of long-tailed classification from the theoretical and practical perspectives of spurious correlation between class-specific and context features in context-poor classes. We are motivated by the observation that the spurious correlation between class-specific and rare context features is more severe than that between class-specific and abundant context features. Breaking the spurious correlation of context-poor classes can effectively improve performance. To tackle this issue, we propose a framework termed spurious correlation of context-poor that can not only mitigate spurious correlation but also more effectively preserve the discriminative class-specific features. Finally, we demonstrate the superiority of the proposed method on four datasets: CIFAR10-LT, CIFAR100-LT, iNaturalist 2018, and ImageNet-LT, where it outperforms recent long-tailed learning algorithms. In the future, we intend to conduct further analyze and explore more effective context learning methods for detection and segmentation.

## 6 Reproducibility Statement

We have made efforts to ensure the reproducibility of our work. All datasets used in our experiments are publicly available, and implementation details are described in Section Experiments of the main text and in Appendix B.1. Meanwhile, to guarantee experimental reproducibility, all experiments were conducted with a fixed random seed. For further reproducibility, we provide the source code of the training and evaluation process as supplementary material. Any theoretical claims made in this work are accompanied by clear assumptions and complete proofs, which can be found in Appendix A.

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

# A    DERIVATION AND PROOF OF FORMULAS

Owing to space limitations, the detailed derivations and proofs of Eq.1 and 2 are provided in this section. As mentioned above in Section Methodology, we expand the $p(y|x)$ to $p(y|f_{cs}, f_{ct}, f_{sc})$, which only assumes the class-specific features $f_{cs}$ is invariant, i.e., $p_{tr}(f_{cs}|y) = p_{te}(f_{cs}|y)$. Based on this, the derivation process of Eq.1 is as follows:

$$
\begin{aligned}
p(y|f_{cs}, f_{ct}, f_{sc}) &= \frac{p(f_{cs}, f_{ct}, f_{sc}|y) \cdot p(y)}{p(f_{cs}, f_{ct}, f_{sc})} \\
&= \frac{p(f_{cs}, f_{ct}|y) \cdot p(f_{sc}|y, f_{cs}, f_{ct}) \cdot p(y)}{p(f_{cs}, f_{ct}, f_{sc})} \\
&= \frac{p(f_{cs}, f_{ct}|y) \cdot p(f_{sc}|y, f_{cs}, f_{ct}) \cdot p(y)}{p(f_{cs}, f_{ct}) \cdot p(f_{sc}|f_{cs}, f_{ct})} \\
&= \frac{p(f_{cs}, f_{ct}|y)}{p(f_{cs}, f_{ct})} \cdot \frac{p(f_{sc}|y, f_{cs}, f_{ct})}{p(f_{sc}|f_{cs}, f_{ct})} \cdot p(y) \\
&= \frac{p(f_{cs}|y) \cdot p(f_{ct}|y, f_{cs})}{p(f_{cs}) \cdot p(f_{ct}|f_{cs}))} \cdot \frac{p(f_{sc}|y, f_{cs}, f_{ct})}{p(f_{sc}|f_{cs}, f_{ct})} \cdot p(y) \\
&= \frac{p(f_{cs}|y)}{p(f_{cs})} \cdot \frac{p(f_{ct}|y, f_{cs})}{p(f_{ct}|f_{cs})} \cdot \frac{p(f_{sc}|y, f_{cs}, f_{ct})}{p(f_{sc}|f_{cs}, f_{ct})} \cdot p(y) \\
&= \underbrace{\frac{p(f_{cs}|y)}{p(f_{cs})}}_{\text{class-specific}} \cdot \underbrace{\frac{p(f_{ct}|y)}{p(f_{ct})}}_{\text{relevant-context}} \cdot \underbrace{\frac{p(f_{ct}|y, f_{cs})}{p(f_{ct}|y)} \cdot \frac{p(f_{ct})}{p(f_{ct}|f_{cs})} \cdot \frac{p(f_{sc}|y, f_{cs}, f_{ct})}{p(f_{sc}|f_{cs}, f_{ct})}}_{\text{spurious correlation}} \cdot p(y)
\end{aligned}
\tag{6}
$$

For a model, we expect that the model can effectively differentiate between class-specific and context features, while utilizing contextual information rather than discarding it entirely. In other words, a well-designed model should be capable of learning not only discriminative features specific to each class but also relevant contextual features that provide supplementary information for improved performance, while avoiding excessive reliance on such contextual features. Therefore, given the class label $y$, the class-specific feature $f_{cs}$ is independent on the context feature $f_{ct}$, we have $p(f_{ct}, f_{cs}|y) = p(f_{ct}|y) \cdot p(f_{cs}|y)$. Meanwhile, from the definition of trivariate conditional probability, we can arrive at

$$
\begin{aligned}
p(f_{ct}|y, f_{cs}) &= \frac{p(f_{ct}, y, f_{cs})}{p(y, f_{cs})} \\
&= \frac{p(y) \cdot p(f_{ct}, f_{cs}|y)}{p(y) \cdot p(f_{cs}|y)} \\
&= \frac{p(f_{ct}, f_{cs}|y)}{p(f_{cs}|y)}
\end{aligned}
\tag{7}
$$

The following formula can be derived:

$$
\begin{aligned}
\frac{p(f_{ct}|y, f_{cs})}{p(f_{ct}|y)} &= \frac{p(f_{ct}, f_{cs}|y)}{p(f_{cs}|y)} \cdot \frac{1}{p(f_{ct}|y)} \\
&= \frac{p(f_{cs}|y) \cdot p(f_{ct}|y)}{p(f_{cs}|y) \cdot p(f_{ct}|y)} \\
&= 1
\end{aligned}
\tag{8}
$$

By substituting Eq.8 into Eq.6, The Eq.6 can be further simplified as follows:

$$
\begin{aligned}
p(y|f_{cs}, f_{ct}, f_{sc}) &= \frac{p(f_{cs}|y)}{p(f_{cs})} \cdot \frac{p(f_{ct}|y)}{p(f_{ct})} \cdot \frac{p(f_{ct}|y, f_{cs})}{p(f_{ct}|y)} \cdot \frac{p(f_{ct})}{p(f_{ct}|f_{cs})} \cdot \frac{p(f_{sc}|y, f_{cs}, f_{ct})}{p(f_{sc}|f_{cs}, f_{ct})} \cdot p(y) \\
&= \frac{p(f_{cs}|y)}{p(f_{cs})} \cdot \frac{p(f_{ct}|y)}{p(f_{ct})} \cdot \frac{p(f_{ct}, f_{cs}|y)}{p(f_{cs}|y) \cdot p(f_{ct}|y)} \cdot \frac{p(f_{ct})}{p(f_{ct}|f_{cs})} \cdot \frac{p(f_{sc}|y, f_{cs}, f_{ct})}{p(f_{sc}|f_{cs}, f_{ct})} \cdot p(y) \\
&= \frac{p(f_{cs}|y)}{p(f_{cs})} \cdot \frac{p(f_{ct}|y)}{p(f_{ct})} \cdot \frac{p(f_{cs}|y) \cdot p(f_{ct}|y)}{p(f_{cs}|y) \cdot p(f_{ct}|y)} \cdot \frac{p(f_{ct})}{p(f_{ct}|f_{cs})} \cdot \frac{p(f_{sc}|y, f_{cs}, f_{ct})}{p(f_{sc}|f_{cs}, f_{ct})} \cdot p(y) \\
&= \underbrace{\frac{p(f_{cs}|y)}{p(f_{cs})}}_{\text{class-specific}} \cdot \underbrace{\frac{p(f_{ct}|y)}{p(f_{ct})}}_{\text{relevant-context}} \cdot \underbrace{\frac{p(f_{ct})}{p(f_{ct}|f_{cs})} \cdot \frac{p(f_{sc}|y, f_{cs}, f_{ct})}{p(f_{sc}|f_{cs}, f_{ct})}}_{\text{spurious correlation}} \cdot p(y).
\end{aligned}
\tag{9}
$$

## B  EXPERIMENTAL RESULTS

### B.1  IMPLEMENTATION DETAILS

We evaluate the proposed method on four widely-used long-tailed datasets: CIFAR10-LT Cao et al. (2019), CIFAR100-LT Cao et al. (2019), iNaturalist 2018 Van Horn et al. (2018), and ImageNet-LT Liu et al. (2019).

**CIFAR10/100-LT**. CIFAR10-LT and CIFAR100-LT are the long-tailed versions, which is formed by sampling from the CIFAR10 and CIFAR100 Krizhevsky & Hinton (2009) datasets with a balanced class distribution. We set the imbalance ratio (i.e., $IR = n_{max}/n_{min}$) to values of 200, 100, 50 for validating SCCP against various levels of imbalance. The $n_{max}$ and $n_{min}$ denote the sample sizes of the most and least frequent classes of the datasets, respectively.

**iNaturalist 2018**. iNaturalist 2018 is a natural long-tailed dataset with an imbalance ratio of 512, it consists of 437.5K training images distributed across 8142 categories.

**ImageNet-LT**. ImageNet-LT is sampled from ImageNet Deng et al. (2009). ImageNet-LT comprises a total of 115.8K training images spanned 1000 classes, resulting in an imbalance ratio of 256.

**Basic Settings**. Follow the standard evaluation metrics, the performances is mainly reported as top-1 classification accuracy. For a more specific analysis, we also provide the experimental results of three partitions: head ($N > 100$), medium ($20 \leq N \leq 100$), and tail ($N < 20$), where $N$ denote the sample numbers of class Liu et al. (2019). For all experiments, the SGD optimizer with a momentum of 0.9 is adopted for all datasets. In CIFAR10-LT and CIFAR100-LT, we use ResNet-32 He et al. (2016) as the backbone network, and train 200 epochs on a single 3090 GPU. The batch size and weight decay are setted as 128 and 5e-4, respectively, and the initial learning rate is setted as 0.1, and is divided by 10 and 5 at the 160th and 180th epochs. For iNaturalist 2018 and ImageNet-LT, the commonly used ResNet-50 He et al. (2016) and ResNext-50 Xie et al. (2017) are utilized as the backbone network, and trained 100 epochs. The learning rate is updated by the cosine scheduler from 0.05 to 0 during training. The batch size is setted as 64. The weight decay of SGD is setted as 1e-4 and 5e-4, respectively. The size of memory bank equals to batch size. For a fair comparison, none of the methods in the experiments are loaded with pre-trained weights. In the tables, the highest-performing results are in bold. $*$ denotes the results quoted from corresponding papers Shao et al. (2024); Zhou et al. (2023), the results of the remained methods are reported from the original paper.

### B.2  DETAILED EXPERIMENTAL RESULTS ON CIFAR100-LT WITH IMBALANCE RATIO 100.

To detailly analyze the results, we also provide the experimental results of three partitions that contain varied numbers of training data: head classes (classes with over 100 samples), medium classes (classes with 20 to 100 samples), and tail classes (classes with fewer than 20 samples) Liu et al. (2019). As shown in Table 4, for the CIFAR100-LT with imbalance ratio 100, SCCP achieves good results 66.9%, 53.5%, and 33.9% on all three groups. Although SCCP did not achieve the best results on the head classes, compared to the second-best method, it showed significant improvements on the medium and tail classes, with improvements of at least 1.9% and 4.2%, respectively. It empirically

Table 4: The top-1 classification accuracy of CIFAR100-LT, the results of different groups (head, medium, and tail classes) is presented. ∗ denotes the results quoted from corresponding published papers, the results of the remained methods are reported from the original paper.

| Dataset | Method | Reference | All | Head | Medium | Tail |
|---------|--------|-----------|-----|------|--------|------|
| CFAR100-LT | CE | | 38.3 | 65.2 | 37.1 | 9.1 |
| | Focal Loss (Lin et al., 2017) | ICCV17 | 38.4 | 65.3 | 38.4 | 8.1 |
| | LDAM∗(Cao et al., 2019) | NeurIPS19 | 42.0 | 61.5 | 41.7 | 20.2 |
| | Decouple(Kang et al., 2019) | ICLR20 | 42.3 | 64.0 | 44.8 | 18.1 |
| | RIDE(Wang et al., 2020) | ICLR20 | 48.0 | 68.1 | 49.2 | 23.9 |
| | CMO (Park et al., 2022) | CVPR22 | 47.2 | 70.4 | 42.5 | 14.4 |
| | SAM (Rangwani et al., 2022) | NeurIPS22 | 45.4 | 64.4 | 46.2 | 20.8 |
| | CUDA(Ahn et al., 2023) | ICLR23 | 47.6 | 67.3 | 50.4 | 21.4 |
| | CSA (Shi et al., 2023) | NeurIPS23 | 46.6 | 64.3 | 49.7 | 18.2 |
| | DiffulT(Shao et al., 2024) | NeurIPS24 | 51.5 | 69.0 | 51.6 | 29.7 |
| | SCCP | Ours | **52.6** | 66.9 | 53.5 | 33.9 |

illustrate that SCCP effectively promotes medium and tail classes without significantly compromising the performance of head classes. The experimental results further proved the effectiveness of our method on context-poor classes.

### B.3 DETAILED EXPERIMENTAL RESULTS ON LARGE-SCALE DATASETS: iNATURALIST 2018, IMAGENET-LT

Similarly, in order to demonstrate the generalization of the proposed method and its impact on different groups, we also report the accuracy of three partitions of classes on large-scale datasets: iNaturalist 2018 and ImageNet-LT, the experimental is presented in Table 5. From the perspective of the overall dataset, SCCP achieves the best performance on both two datasets. However, it do not achieve optimal results across all three groups. As shown in the experimental results of Table 5, on the large-scale datasets iNaturalist 2018 and imagenet-LT, our approach demonstrates the effectiveness on the medium and tail classes (classes with fewer than 100 samples). In contrast to head classes that contain thousands of samples, medium and tail classes are particularly susceptible to spurious correlation. Therefore, as evidenced by the results presented in Table 5, our method is capable of extracting the discriminative class-specific information while mitigating the spurious correlation between class-specific and context features effectively.

### B.4 LOSS LANDSCAPES VISUALIZATION

To further illustrate the model's ability to break the spurious correlation between context and class-specific features, the Figure 7 presents the loss landscapes visualizations Li et al. (2018); Rangwani et al. (2022) of head, medium, and tail classes on CIFAR100 with long-tail ratio 100 in BCL and SCCP. From the first row of the Figure 7, we can observed that the loss surface of BCL is relatively low but highly sharp, suggesting that the model suffers from poor generalization performance and is prone to overfitting. In contrast, the proposed method demonstrates lower loss values along with a more gradual and flat minima, indicating superior stability and generalization capability. Therefore, SCCP enables both the mitigation of saddle point entrapment and the enhancement of representational generalization. As shown in the second row of Figure 7, the loss surfaces of SCCP is notably flatter and smoother in both head, medium, and tail classes, which further supports the conclusion that the method effectively mitigates the spurious correlation of context-poor classes between context features and the class-specific features.

### B.5 HESSIAN SPECTRAL ANALYSIS

Figure 8 presents the class-specific hessian spectral analysis Dinh et al. (2017); Dauphin et al. (2014); Rangwani et al. (2022) of CE, BCL, and SCCP on CIFAR100 with long-tail ratio 100, respectively. Columns 1−2, 3−4, and 5−6 in the figure illustrate the hessian spectral analysis conducted for each class under the CE, BCL, and SCCP, respectively. A comparison of the cor-

Table 5: The top-1 classification accuracy of iNaturalist 2018 and Imagenet-LT, the results of different groups (head, medium, and tail classes) is presented. ∗ denotes the results quoted from corresponding published papers, the results of the remained methods are reported from the original paper.

| Dataset | Method | Reference | All | Head | Medium | Tail |
|---------|--------|-----------|-----|------|--------|------|
| iNa-18 | CE ∗ | | 61.2 | 72.2 | 63.0 | 57.2 |
| | LDAM∗(Cao et al., 2019) | NeurIPS19 | 70.1 | 64.1 | 70.5 | 71.2 |
| | Decouple-cRT (Kang et al., 2019) | ICLR20 | 68.2 | 73.2 | 68.8 | 66.1 |
| | Decouple-LWS (Kang et al., 2019) | ICLR20 | 69.5 | 71.0 | 69.8 | 68.8 |
| | MiSLAS∗ (Zhong et al., 2021) | CVPR21 | 71.6 | 73.2 | 72.4 | 70.4 |
| | CE + CMO (Park et al., 2022) | CVPR22 | 68.9 | 76.9 | 69.3 | 66.6 |
| | CE-DRW + CMO (Park et al., 2022) | CVPR22 | 70.9 | 68.2 | 70.2 | 72.2 |
| | LDAM-DRW + CMO (Park et al., 2022) | CVPR22 | 69.1 | 75.3 | 69.5 | 67.3 |
| | BS + CMO (Park et al., 2022) | CVPR22 | 70.9 | 68.8 | 70.0 | 72.3 |
| | CE-DRW + LAS + CMO(Park et al., 2022) | CVPR22 | 71.8 | 69.6 | 72.1 | 71.9 |
| | GCL (Mengke Li, 2022) | CVPR22 | 71.5 | 66.4 | 71.7 | 72.5 |
| | MiSLAS + PLOT (Zhou et al., 2023) | ICLR24 | 72.1 | 73.1 | 72.9 | 71.2 |
| | cRT + Mixup + PLOT(Zhou et al., 2023) | ICLR24 | 71.3 | 74.2 | 72.5 | 69.4 |
| | DR+H2T (Li et al., 2024b) | AAAI24 | 71.8 | 71.7 | 72.3 | 71.3 |
| | MisLAS+H2T (Li et al., 2024b) | AAAI24 | 72.1 | 69.7 | 72.5 | 72.2 |
| | GCL+H2T (Li et al., 2024b) | AAAI24 | 71.6 | 67.7 | 71.9 | 72.2 |
| | SAM (Li et al., 2025) | ICML25 | 71.8 | 68.4 | 72.0 | 72.5 |
| | SCCP | Ours | **72.7** | 73.3 | 73.2 | 71.9 |
| Ima-LT | CE ∗ | | 44.4 | 65.9 | 37.5 | 7.7 |
| | LDAM∗(Cao et al., 2019) | NeurIPS19 | 52.9 | 63.0 | 50.5 | 35.5 |
| | Decouple∗ (Kang et al., 2019) | ICLR20 | 46.7 | 56.6 | 44.2 | 27.4 |
| | BS∗ (Ren et al., 2020) | NeurIPS20 | 52.3 | 64.1 | 48.2 | 33.4 |
| | LADE∗ (Hong et al., 2021) | CVPR21 | 52.3 | 64.4 | 47.7 | 34.3 |
| | RSG∗ (Wang et al., 2021a) | CVPR21 | 51.8 | 63.2 | 48.2 | 32.2 |
| | DisAlign∗ (Zhang et al., 2021) | CVPR21 | 52.9 | 61.3 | 52.2 | 31.4 |
| | MiSLAS∗ (Zhong et al., 2021) | CVPR21 | 52.7 | 61.7 | 51.3 | 35.8 |
| | GCL (Mengke Li, 2022) | CVPR22 | 54.5 | 62.2 | 48.6 | 52.1 |
| | ResLT (Cui et al., 2022) | TPAMI22 | 52.9 | 63.0 | 50.5 | 35.5 |
| | BCL (Zhu et al., 2022) | CVPR22 | 56.0 | 65.7 | 53.7 | 37.3 |
| | CSA (Shi et al., 2023) | NeurIPS23 | 49.7 | 63.6 | 47.0 | 23.8 |
| | AREA (Chen et al., 2023) | ICCV23 | 49.5 | 58.7 | 46.8 | 29.1 |
| | PLOT (Zhou et al., 2023) | ICLR24 | 53.5 | 61.4 | 52.3 | 37.5 |
| | H2T (Li et al., 2024b) | AAAI24 | 54.6 | 62.4 | 48.8 | 52.2 |
| | DiffuLT(Shao et al., 2024) | NeurIPS24 | 56.4 | 63.3 | 55.6 | 39.4 |
| | DisA(Gao et al., 2024) | ICML24 | 54.5 | 65.0 | 52.1 | 33.0 |
| | SAM (Li et al., 2025) | ICML25 | 54.3 | 63.9 | 52.2 | 34.4 |
| | SCCP | Ours | **57.0** | 67.5 | 54.9 | 37.9 |

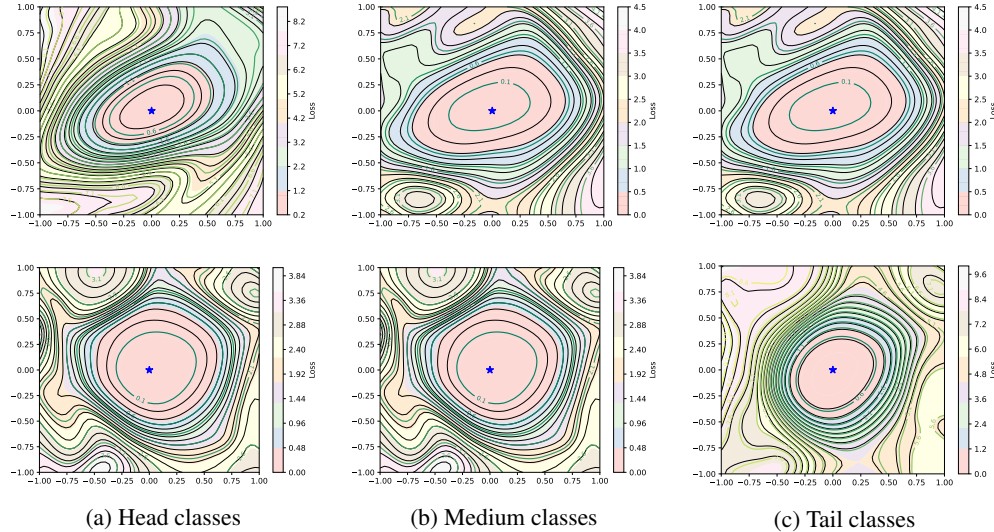

| (a) Head classes | (b) Medium classes | (c) Tail classes |

Figure 7: Loss landscapes of head, medium, and tail classes on CIFAR100 with imbalance ratio 100 in BCL and SCCP.

responding subgraphs across the three figures reveals that our method exhibits significantly better performance in terms of both the largest and smallest eigenvalues ($\lambda_{max}$ and $\lambda_{min}$) compared to BCL and CE. Meanwhile, we also conduct hessian spectral analyses on the entire dataset as well as on the head, medium, and tail classes separately. The experimental results are presented in Figure 9. As evidenced by the results, the SCCP demonstrates a significant advantage over the other two algorithms. By reducing the likelihood of convergence to saddle points, the proposed method facilitates the development of representations with improved generalization capability.

## B.6 SUPPLEMENTARY ABLATION EXPERIMENTS

**Strategies of generating memory bank**. We implement different strategies of generating memory bank to better analyze our proposed method, the experimental results are presented in Table 6. We believe that the memory bank should retain high-quality samples and maintain representation from different categories to enhance contextual diversity. To evaluate this design, we conduct experiments using the following alternative strategies: (1) the memory bank is reset at the beginning of each epoch; (2) samples from the previous batch are used directly as the memory bank; (3) categories containing only one sample in the memory bank are excluded; and (4) categories with only one sample are not retained, and the memory bank is reinitialized at the start of each epoch. Compared with (1)-(4), SCCP has basically improved by 0% to 1.4% on the two datasets across various imbalance ratio, and the improvement becomes more obvious as the imbalance ratio increases, which indicates that SCCP is more effective in breaking spurious correlation.

Table 6: Ablation experiments to evaluate the effects of generating different memory bank.

| Method | CIFAR100 | | | CIFAR10 | | |
|---|---|---|---|---|---|---|
| | 200 | 100 | 50 | 200 | 100 | 50 |
| (1) | 47.7 | 52.5 | 56.3 | 81.7 | 85.0 | 87.6 |
| (2) | 47.3 | 52.2 | 56.7 | **82.0** | 84.8 | 87.6 |
| (3) | 47.6 | 52.3 | 56.1 | 81.4 | 85.0 | 87.7 |
| (4) | 46.4 | 51.5 | 56.5 | 81.9 | 85.3 | 87.3 |
| SCCP | **47.8** | **52.6** | **56.8** | **82.0** | **85.5** | **87.9** |

**Supplementary Experiments of Hyper-parameters.** We conduct a experiment to evaluate the influence of hyper-parameters in this method. Table 7 presents the effect of the hyper-parameters in foreground-background distribution mask. We adjust the threshold of the distribution mask.

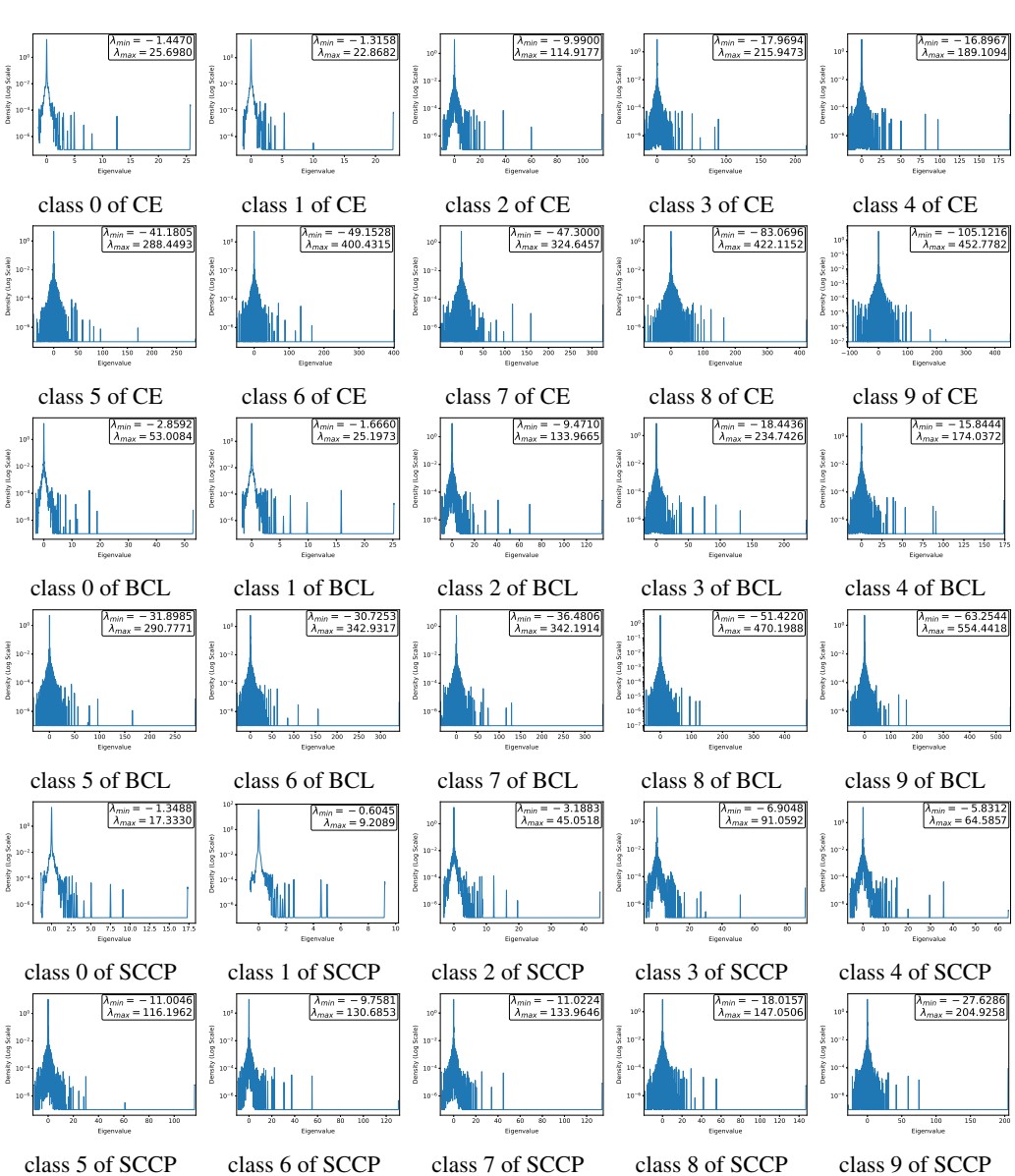

Figure 8: Hessian spectrum analysis of the CE, BCL, and SCCP on CIFAR100 with imbalance ratio 50.

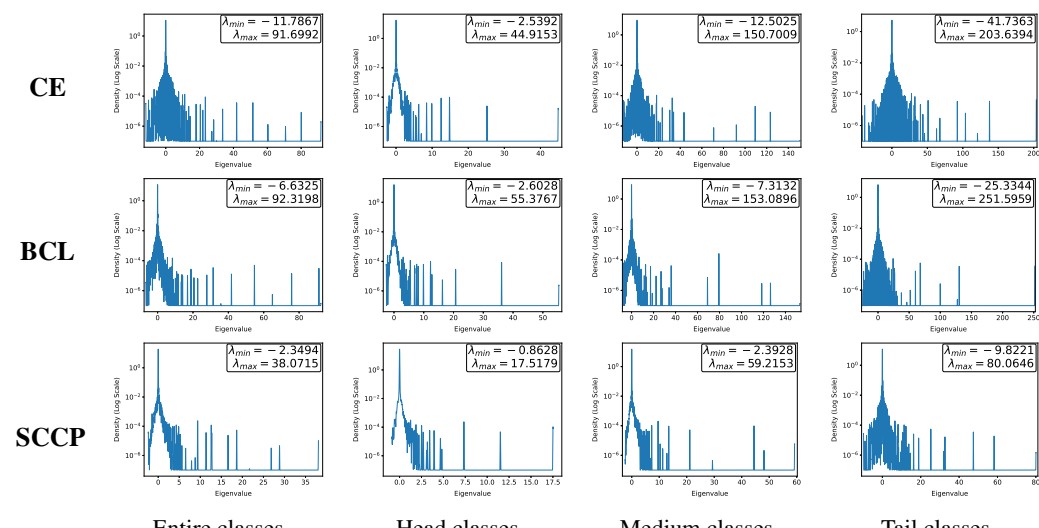

CE

BCL

SCCP

| Entire classes | Head classes | Medium classes | Tail classes |

Figure 9: Hessian spectrum analysis of head, medium, and tail classes on CIFAR100 with imbalance ratio 100.

Table 7: Evaluation of threshold effects in the foreground-background distribution mask.

| Threshold | CIFAR100 | | | CIFAR10 | | |
|---|---|---|---|---|---|---|
| | 200 | 100 | 50 | 200 | 100 | 50 |
| 0.1 | 47.6 | 51.7 | **56.8** | 81.9 | 85.0 | 87.3 |
| 0.2 | 47.3 | 51.7 | **56.8** | 81.5 | 85.3 | 87.5 |
| 0.3 | 47.7 | 51.6 | 56.7 | 81.8 | 85.0 | 87.5 |
| 0.4 | 47.6 | 52.2 | 56.7 | 81.9 | 85.4 | 87.8 |
| 0.6 | 47.7 | 52.3 | 56.3 | 81.3 | 85.3 | 87.7 |
| 0.7 | 47.7 | 52.3 | 56.2 | 81.1 | 85.2 | 87.7 |
| 0.8 | 47.3 | 52.1 | 56.7 | 81.9 | 85.1 | 87.6 |
| 0.5 | **47.8** | **52.6** | **56.8** | **82.0** | **85.5** | **87.9** |

Through iterative adjustments, we find that the optimal performance is achieved with 47.8%, 52.6%, 56.8%, 82.0%, 85.5%, 87.9% spanning various imbalance ratios, when the threshold is equal to 0.5.

Similarly, to validate the influence of the memory size on the memory bank, we further conduct experimental analysis, the results are shown in Table 8. The best setting for memory size is determined to equal the batch size. The Table 7 and 8 show that, as the training process progresses, most samples will obtain an excellent distribution mask, so the two parameters have little influence on the experimental results, which also indicates the robustness of our method. Consequently, we establish 0.5 and batch size for the two parameters as the default settings for the proposed method.

Table 8: Ablation experiments to evaluate the effect of size in the memory bank.

| Size | CIFAR100 | | | CIFAR10 | | |
|---|---|---|---|---|---|---|
| | 200 | 100 | 50 | 200 | 100 | 50 |
| 256 | 47.7 | 52.6 | 56.4 | 81.6 | 84.8 | 87.4 |
| 384 | 47.3 | 52.5 | 56.6 | 81.8 | 84.9 | 87.5 |
| 512 | 47.5 | 52.0 | 56.6 | 81.9 | 84.9 | 87.4 |
| 128 | **47.8** | **52.6** | **56.8** | **82.0** | **85.5** | **87.9** |

## C  RELATED WORK

Most existing long-tailed methods are used to re-balance the long-tailed data for improving the classification performance can be categorized as 1) re-sampling and re-weighting Wei et al. (2022); Shi et al. (2023); Cao et al. (2019); Cui et al. (2019) and 2) data augmentation Li et al. (2024b); Park et al. (2022); Shao et al. (2024); Mengke Li (2022).

Resampling methods mitigate class imbalance by either replicating samples from tail classes or discarding samples from head classes based on specific criteria. However, these metohds may lead to overfitting on scarce tail samples or the elimination of potentially informative head classes data. Re-weighting techniques attempt to achieve more balanced predictions by assigning different weights within loss functions, yet on large-scale long-tailed datasets, such approaches can substantially increase optimization complexity. For instance, for re-balancing class priors and promoting more separable representations, Wei et al. proposed the Open-sampling, which utilizes open-set noisy labels to re-balance class priors, where each open-set instance is assigned a label sampled from a distribution complementary to the original class priors Wei et al. (2022). Conversely, Cao et al. examined the margins of training samples and introduced a label-distribution-aware margin loss, designed to encourage larger margins for minority classes Cao et al. (2019).

Data augmentation methods are generally categorized into two types: those that independently enrich all samples to improve head and tail categories at the same time and those that transfer shared features from head classes to tail classes, thereby effectively improving the performance of the tail classes without reducing the performance of the head classes. Li et al. introduced a plug-and-play module that replaces part of the tail classes feature representation with that of the head classes Li et al. (2024b). This feature fusion strategy substantially enriches the representational diversity of tail classes. Shao et al. utilized a diffusion model to generate approximately in-distribution samples Shao et al. (2024). Although these samples slightly deviate from the real data distribution and contain blended category information, they serve as valuable training instances that improve generative models in long-tailed classification tasks. Nevertheless, the high computational cost of this method makes it impractical for large-scale long-tailed datasets. Park et al. attributed the inferior performance of tail classes to their poor generalization capability Park et al. (2022). To overcome this issue, they enhance the sample diversity by injecting informative contextual backgrounds from head classes into tail classes.

Recently, some scholars have discussed the influence of invariant and variable features on feature representation. While these approaches can still be broadly categorized into the aforementioned two groups, they offer a novel perspective for the long-tail problem by providing theoretical interpretations Qi et al. (2022); Yi et al. (2022); Tang et al. (2022); Park et al. (2022); Shi et al. (2023). Qi et al. assumed that the context is also invariant to class; they consider the classes as the varying environments to resolve context bias while assuring this similarity to be invariant across all classes Qi et al. (2022). Tang et al. posited that the perious study over-emphasize the class distribution while neglecting to learn attribute-invariant features. Therefore, they introduce a novel research problem: generalized long-tailed classification, meanwhile an invariant feature learning method is proposed as the first strong baseline for generalized long-tailed Tang et al. (2022). Shi et al. systematically re-examined re-sampling methods and demonstrated that such methods can substantially improve model generalization, provided that the training images are free from semantically irrelevant contextual cues. Building upon this finding, they further introduced a novel context-shift augmentation module to enhance the representation learning of tail classes Shi et al. (2023).

Different from the algorithms mentioned above, we posit that both class-specific and context features contribute to enhancing the model's classification performance. However, spurious correlation arising from an overreliance on context features will degrade overall model performance. We conduct theoretical analysis and empirical investigation revealing that spurious correlation between class-specific features and rare context features in context-poor classes can substantially impair the performance of tail classes. By effectively alleviating the spurious correlation in context-poor classes, the proposed method greatly improves the models generalization ability and robustness.

## D   THE SUPPORT FROM LARGE LANGUAGE MODELS

This work benefited from the support of large language models, which assisted in the manuscript writing.

