# OpenReview forum: "Class-specific Feature Learning through Mitigating Spurious Correlation in Context-Poor Classes for Long-Tailed Classification"
_ICLR.cc/2026/Conference — ICLR 2026 Conference Withdrawn Submission_

### Official Review · Reviewer_rXQG · 2025-10-26

**Soundness:** 3
**Presentation:** 2
**Contribution:** 2
**Rating:** 4
**Confidence:** 3

**Summary:**

This work addresses the challenge of long-tailed data in real-world scenarios, where models tend to favor context-rich classes and neglect context-poor ones. The proposed framework, Spurious Correlation of Context-Poor (SCCP), progressively breaks spurious correlations by using Grad-CAM-based context separation, sample generation, and contrastive learning with refined class centers. Experimental results verify the effectivenss of proposed methods.

**Strengths:**

(1) Exploring context rich and context poor features for imbalanced learning is important.

**Weaknesses:**

(1) The writing is confusing. For example, what is the meaning of $x_1$, $x_2$, $x_3$, and $x_4$ in the Fig.3 ? Also what is the meaning of $O_1$, $O_2$, $O_3$, and $O_4$? I recommend the author to write the method section more clearly.

(2) The visualization results are interesting, but do not prove the effectiveness of the method. For example, the author should visualize the Grad-CAM before and after applying your method. In addition, I think the visualization result in Fig.1  (b) do not verify ``SCCP primarily emphasizes class-specific features".  I don't think this image contains any context-rich information.

(3) Actually, context rich has been explored in previous works [a]. What the differences between SCCP and these? In addition, CAM is also been explored in imbalacned learning [b].

[a] Park, Seulki, et al. "The majority can help the minority: Context-rich minority oversampling for long-tailed classification." Proceedings of the IEEE/CVF conference on computer vision and pattern recognition. 2022.

[b] Chu, Peng, et al. "Feature space augmentation for long-tailed data." European conference on computer vision. Cham: Springer International Publishing, 2020.

**Questions:**

see weakness

---

### Official Review · Reviewer_ua83 · 2025-11-01

**Soundness:** 2
**Presentation:** 3
**Contribution:** 1
**Rating:** 2
**Confidence:** 4

**Summary:**

This paper studies learning from long-tailed data, and proposes a new perspective of context imbalance. It first explains that models tend to perform well on the context-rich classes, while underperforms on context-poor classes, leading to the spurious
correlation between class-specific and context-poor features. To address this problem, the authors propose to focus on class-specific features to avoid spurious correlations. The proposed method consists of a Grad-CAM-based segmentation module and a contrastive loss based on more reasonable class centers. Experiments on multiple long-tailed datasets demonstrate the effectiveness of the proposed method.

**Strengths:**

- The paper is easy to follow. The authors provide proper illustrations, formulations and explanations for better understanding.
- The experimental results is better than the compared methods.
- The code is provided for reproducibility.

**Weaknesses:**

- The main claim of this paper, i.e., context imbalance, is not demonstrated via any empirical study. Therefore, the motivation of this paper is not convincing enough.
- The novelty is limited. Mitigating spurious correlation using Grad-CAM is similar to the previous work [1][2]. Leveraing class centers for improving contrastive loss is also similar to previous works [3].
- In section 3.2, the authors claim that sample with high-confidence scores are not suitable. However, there is no theoretical or empirical evidence. The upper threshold is set as 0.8, which however, is not analyzed in the experiments. The experiment in Table 7 only analyzes the threshold of 0.5.
- The numbers in the second line of Table 1 (e.g. 200, 100, 50, 256, 512) is not explained. The authors should annotate "Imbalance Ratio" for these numbers. Table 3 should also be modified.

[1] Bag of Tricks for Long-Tailed Visual Recognition with Deep Convolutional Neural Networks.

[2] How Re-sampling Helps for Long-Tail Learning?

[3] Improving Tail-Class Representation with Centroid Contrastive Learning.

[4] Balanced Contrastive Learning for Long-Tailed Visual Recognition.

**Questions:**

- Experimental results show that the performance even surpasses Diffusion-based method (DiffuLT) [1]. Does the proposed method leverages any extra models or knowledges? Or is there any explanations?

[1] DiffuLT: Diffusion for Long-tail Recognition Without External Knowledge.

---

### Official Review · Reviewer_9etT · 2025-11-01

**Soundness:** 2
**Presentation:** 2
**Contribution:** 2
**Rating:** 4
**Confidence:** 3

**Summary:**

This paper investigates long-tailed classification from the perspective of spurious correlation between class-specific and context features in context-poor classes (tail and medium classes). The key insight is that spurious correlation between class-specific features and rare context features is more severe in context-poor classes due to insufficient contextual diversity. The authors propose the SCCP framework to mitigate this issue through: (1) Breaking Spurious Correlation (BSC) using Grad-CAM to segment context and foreground regions, generating synthetic samples with varied contexts, and (2) applying Logit Adjustment (LA) loss on generated samples to enhance discriminative class-specific feature learning. The method achieves 52.6% accuracy on CIFAR100-LT (IR 100), 72.7% on iNaturalist 2018, and 57.0% on ImageNet-LT, demonstrating improvements particularly on tail classes (33.9% on CIFAR100-LT tail classes vs. 30.4% baseline).

**Strengths:**

**Novel Perspective on Long-Tail Problem**: Focuses on spurious correlation between class-specific and context features as the root cause of tail class failure, rather than just class imbalance. The Bayesian derivation (Equation 9) explicitly identifies the spurious correlation term. Empirical observation that spurious correlation worsens with increasing imbalance ratio (Figure 1 shows overfitting increases from IR 50→100→200) provides clear motivation. The context-poor vs. context-rich class distinction offers a new lens for understanding long-tail dynamics.

**Strong Empirical Results on Tail Classes**: Achieves best or competitive performance across four datasets. Particularly strong on tail classes: CIFAR100-LT tail improves from 30.4%→33.9% (+11.5% relative), while maintaining head class performance (66.9% vs. 65.2% baseline). On ImageNet-LT and iNaturalist 2018, it shows +0.6% over the second-best methods. Loss landscape visualization (Figure 5) shows flatter, more generalizable minima compared to the BCL baseline.

**Weaknesses:**

**Method Complexity with Unclear Details**: Multiple loosely connected components: Grad-CAM segmentation (missing layer choice, confidence threshold, failure handling for ViTs), synthetic sample generation (no algorithm provided, label noise rate unknown), "reasonable class centers" (computation method unspecified), and contrastive loss + LA loss (interaction unclear).

**Weak Theoretical Connection**: Equation 9 identifies s spurious correlation term, but the paper does not prove that BSC reduces it. The connection between Grad-CAM segmentation → mask generation → contrastive learning and the theoretical term is hand-wavy. No guarantees on when the method works/fails or how Grad-CAM quality affects performance. Method appears to be effective data augmentation + contrastive learning repackaged as spurious correlation mitigation without rigorous justification.

**Questions:**

Please refer to the Weakness

---

### Official Review · Reviewer_7orV · 2025-11-02

**Soundness:** 2
**Presentation:** 2
**Contribution:** 2
**Rating:** 2
**Confidence:** 4

**Summary:**

This paper tackles the long-tailed recognition problem, where head classes dominate the training set and tail classes are under-represented. The authors argue that tail-class images not only suffer from few samples but also from spurious correlations between object and background, that is, tail classes appear in limited contexts, causing the model to rely on background instead of real object semantics. To address this, the paper proposes a Class-specific Feature Learning (CSFL) framework that explicitly breaks spurious correlations by: 1) Using Grad-CAM to separate foreground (object) and background (context); 2) Maintaining a feature bank that stores high-confidence background samples from head classes; 3) Generate new training samples by replacing the background of context-poor tail samples with diverse backgrounds from the bank; 4) Refining class prototypes through contrastive learning, ensuring features of the same class cluster together regardless of background. Experiments on CIFAR-LT, ImageNet-LT, and iNaturalist show improved tail performance compared to re-weighting and logit-adjustment baselines.

**Strengths:**

- The paper identifies a non-trivial and intuitive failure reason of long-tailed recognition: tail classes are not only few but also context-limited.
- The idea of increasing context via Grad-CAM-guided background replacement is novel and easy to understand.
- The method is simple to implement and can be combined with standard long-tail training objectives.
- Extensive experiments demonstrate consistent improvements on multiple benchmarks.

**Weaknesses:**

- The writing of this paper can be further improved.. For example, “long-tail distribution will inevitably in real world” — incorrect grammar;
- Some expressions are not smooth and consistent in logic, for example, "The validity of this formula relies on the strong assumption that ptr(x|y) = pte(x|y). However, this assumption cannot be guaranteed in practical applications." What do you want to tell? If this assumption does not satisfy, what is the next? This paragraph just ended here.
- Missing parentheses around all citations in the paper (e.g., “is expensive Zhang et al. (2023)”), should use \citep instead of \cite, I think.
- The missed explanation of some notations, for example, the LA loss in equation  (5). Although the related paper is cited, readers may not get the idea without checking the related works. At least, a brief explanation is needed.
- The entire pipeline depends on Grad-CAM to separate foreground and background. The paper briefly mentions filtering by confidence score but does not quantify how often Grad-CAM masks are accurate. Can the authors provide quantitative evidence (e.g., foreground IoU with GT masks) showing that Grad-CAM segmentation is reliable for tail classes? Besides, are there any other options to replace Grad-CAM, for example, the SAM-based models in recent two years. Grad-CAM is pretty old considering the time.
- The proposed context replacement is very similar to several prior works [1,2,3]. What is the key difference beyond using Grad-CAM for masking?
- The class centers refinement step is under-explained. For example, how often are class centers updated, every iteration or epoch? Does the class center change differently for head vs. tail classes after augmentation?

[1] The Majority Can Help The Minority: Context-rich Minority Oversampling for Long-tailed Classification

[2] MetaSAug: Meta Semantic Augmentation for Long-Tailed Visual Recognition

[3] Feature Fusion from Head to Tail for Long-Tailed Visual Recognition

**Questions:**

- The method assumes that giving tail classes more diverse backgrounds will always help, but this may not hold true when the background is actually part of the class identity. For example, “penguins” usually appear in snow. Replacing such meaningful contexts might confuse the model instead of helping. Discussing when context replacement could hurt performance would make the paper more complete.

---

### Note · Authors · 2025-11-12

I have read and agree with the venue's withdrawal policy on behalf of myself and my co-authors.